# Effect of Moderate to Severe Hepatic Steatosis on Vaccine Immunogenicity against Wild-Type and Mutant Virus and COVID-19 Infection among BNT162b2 Recipients

**DOI:** 10.3390/vaccines11030497

**Published:** 2023-02-21

**Authors:** Ka Shing Cheung, Lok Ka Lam, Xianhua Mao, Jing Tong Tan, Poh Hwa Ooi, Ruiqi Zhang, Kwok Hung Chan, Ivan F. N. Hung, Wai Kay Seto, Man Fung Yuen

**Affiliations:** 1Department of Medicine, School of Clinical Medicine, The University of Hong Kong, Queen Mary Hospital, Hong Kong; 2Department of Medicine, The University of Hong Kong, Shenzhen Hospital, Shenzhen 518009, China; 3Department of Microbiology, The University of Hong Kong, Queen Mary Hospital, Hong Kong; 4State Key Laboratory of Liver Research, The University of Hong Kong, Hong Kong

**Keywords:** COVID-19, SARS-CoV-2, vaccination, NAFLD, cirrhosis

## Abstract

Background: We aimed to investigate the effect of non-alcoholic fatty liver disease (NAFLD) on BNT162b2 immunogenicity against wild-type SARS-CoV-2 and variants and infection outcome, as data are lacking. Methods: Recipients of two doses of BNT162b2 were prospectively recruited. Outcomes of interest were seroconversion of neutralizing antibody by live virus microneutralization (vMN) to SARS-CoV-2 strains (wild-type, delta and omicron variants) at day 21, 56 and 180 after first dose. Exposure of interest was moderate-to-severe NAFLD (controlled attenuation parameter ≥ 268 dB/M on transient elastography). We calculated adjusted odds ratio (aOR) of infection with NAFLD by adjusting for age, sex, overweight/obesity, diabetes and antibiotic use. Results: Of 259 BNT162b2 recipients (90 (34.7%) male; median age: 50.8 years (IQR: 43.6–57.8)), 68 (26.3%) had NAFLD. For wild type, there was no difference in seroconversion rate between NAFLD and control groups at day 21 (72.1% vs. 77.0%; *p* = 0.42), day 56 (100% vs. 100%) and day 180 (100% and 97.2%; *p* = 0.22), respectively. For the delta variant, there was no difference also at day 21 (25.0% vs. 29.5%; *p* = 0.70), day 56 (100% vs. 98.4%; *p* = 0.57) and day 180 (89.5% vs. 93.3%; *p* = 0.58), respectively. For the omicron variant, none achieved seroconversion at day 21 and 180. At day 56, there was no difference in seroconversion rate (15.0% vs. 18.0%; *p* = 0.76). NAFLD was not an independent risk factor of infection (aOR: 1.50; 95% CI: 0.68–3.24). Conclusions: NAFLD patients receiving two doses of BNT162b2 had good immunogenicity to wild-type SARS-CoV-2 and the delta variant but not the omicron variant, and they were not at higher risk of infection compared with controls.

## 1. Background

Coronavirus disease 2019 (COVID-19), the illness caused by severe acute respiratory syndrome coronavirus 2 (SARS-CoV-2), emerged in late 2019 and remains a public health burden globally. As of January 2023, there have been more than 600 million confirmed cases of COVID-19, including over 6 million deaths, reported to the World Health Organization. Measures to dampen the spread of COVID-19 have been of paramount importance to avoid the breakdown of major healthcare systems and to reduce excess mortality during peak infection periods. Vaccination is considered to be the most promising approach in preventing and reducing infection, severe disease and death [1].

Underlying comorbidities, including hypertension [2], diabetes mellitus (DM) [3] and obesity [4], have been shown to be associated with adverse COVID-19 outcomes and lower vaccine immunogenicity [5,6]. Chronic liver disease confers a higher risk of infection and disease severity of COVID-19, particularly those with liver cirrhosis and liver transplantation [7,8]. In addition, occurrence of liver injury in patients is associated with prolonged hospitalization [9]. A systemic meta-analysis by Wong et al. revealed that liver injury is mostly associated with severe forms of COVID-19 [10]. Obesity-associated inflammation is a risk factor for non-alcoholic fatty liver disease [11] and is associated with an increased risk of complications in COVID-19 patients [12].

With a prevalence of 32% worldwide for non-alcoholic fatty liver disease (NAFLD) [13], concerns have also been raised about the response to COVID-19 vaccination in this population. Wang et al. [8] reported that BBIBP-CorV (inactivated vaccine) was safe, with good immunogenicity (95.5% had detectable levels of neutralizing antibody after two doses of vaccine). This study, however, did not recruit patients without NAFLD for comparison. While the effect of moderate to severe hepatic steatosis on the BNT162b2 vaccine immunogenicity in NAFLD patients was recently studied [14], data on neutralizing antibodies against mutant viruses, for instance, delta or omicron, and data on long-term immunogenicity (e.g., 6 months) and infection outcome are lacking. We aimed to further evaluate the vaccine immunogenicity (in term of neutralizing antibody response) and vaccine protection from COVID-19 infection in NAFLD subjects receiving the BNT162b2 vaccine in comparison with non-NAFLD subjects.

## 2. Methods

### 2.1. Study Design

This is a prospective cohort study recruiting adult BNT162b2 vaccine recipients from two vaccination centers (Sun Yat Sen Memorial Park Sports Centre and Queen Mary Hospital) in Hong Kong. Exclusion criteria included age less than 18 years, organ transplant or blood transplant, in receipt of immunosuppressives or chemotherapy, other medical diseases (malignancy, hematological, rheumatological and autoimmune diseases), as well as prior COVID-19 infection (identified from both history taking and presence of antibodies to SARS-CoV-2 nucleocapsid (N) protein).

Study subjects received two doses of BNT162b2 (0.3 mL) intramuscularly 3 weeks apart. Their blood samples were collected at four timepoints: (i) before vaccination (baseline), (ii) 21 days after the first dose, (iii) 56 days after the first dose and (iv) 180 days after the first dose.

SARS-CoV-2 infection can be inhibited by blocking viral entry by inducing anti-SARS-CoV-2 neutralizing antibodies, namely receptor-binding domain (RBD) and N-terminal domain (NTD) of the spike protein [15]. Several methods are applied to evaluate the antibody level, such as immunofluorescence (IF), enzyme-linked immunoassay (ELISA) and live virus microneutralization (vMN) assay. IF and ELISA detect antibodies that can bind to virus or viral antigen, while vMN assay measures the neutralizing activity against the virus at the protein expression level. Anti-RBD antibody, which is evaluated by an ELISA-based surrogate neutralizing antibody (sNAb) test [16], is commonly used to express COVID-19 vaccine immunogenicity. On the other hand, vMN results express the total neutralizing activity, including anti-RBD and anti-NTD antibodies. Viral neutralization tests (VNTs) are regarded as the gold standard for serological detection [17], as vMN results indicate inactivation of infectious virus. VNTs, which are strongly correlated with disease protection, were chosen as the indicator of COVID-19 vaccine efficacy in our study.

vMN assay was carried out in 96-well plate where serum samples were diluted in 2 folds serially starting from 1:10 (Gibco, Green Island, NY, USA). Diluted serum was mixed with 100 TCID50 (50% tissue culture infective dose) of SARS-CoV-2 and incubated at 37 degrees Celsius for one hour. The mixture was merged with VeroE6 cells and incubated at 37 degrees Celsius and 5% carbon dioxide. After incubation for 72 h, the cytopathic effect was evaluated by examination under inversion microscopy. With reference to the standardization for SARVS-CoV-2 human immunoglobulin by the World Health Organization’s International Standard, the titer of vMN antibody was adopted from the highest dilution with 50% inhibition of cytopathic effect. vMN positivity indicates seroconversion and was defined as a titer equal to or greater than 10 (31.25 IU/mL). vMN titers of three different strains of COVID-19—wild type, delta variants and omicron BA.1 variants—were measured respectively.

The study was approved by the Institutional Review Board of the University of Hong Kong (HKU) and Hong Kong West Cluster (HKWC) of Hospital Authority.

### 2.2. Outcome of Interest

Primary outcomes of interest were seroconversion rate at three time points (day 21, day 56 and day 180 after first dose of vaccination) to three different strains of SARS-CoV-2: wild type, delta variants and omicron variants.

Secondary outcomes of interest were (i) COVID-19 infection rate and (ii) overall and individual adverse reactions. For the outcome of infection, subjects were followed until 18 May 2022 (study end date). COVID-19 was confirmed by either Rapid Antigen test (RAT) or Deep Throat Saliva (DTS). For the outcome of adverse reactions, subjects were requested to report any adverse reactions daily for 7 days after each dose of vaccine. They were classified into local reactions (pain, erythema, swelling and itchiness) and systemic reactions (fever, chills, headache, fatigue, myalgia, arthralgia, nausea, vomiting, diarrhea, skin rash and facial drooping). The severity of each adverse reaction was graded as 1 (mild), 2 (moderate), 3 (severe) and 4 (potentially life-threatening disease), with reference to the toxicity grading scale by the United States Department of Health and Human Services (HHS) [18].

### 2.3. Exposure of Interest

Controlled Attenuation Parameter measured by transient elastography (TE) using Fibroscan (Echosens, Paris, France) was used to define the presence of hepatic steatosis, which was further classified into different severity: mild (CAP 248–267 dB/m), moderate (CAP 268–279 dB/m) and severe (CAP 280 dB/m) [19].

Subjects with moderate to severe hepatic steatosis (i.e., CAP ≥ 268 dB/M) were grouped as “NAFLD” and those with mild or no hepatic steatosis were grouped as control. This is because subjects with moderate or severe hepatic steatosis have markedly higher risks in various clinical outcomes, including fibrosis, HCC and cardiovascular diseases, than those with mild hepatic steatosis [20,21]. Covariates included age, sex, overweight/obesity [22], diabetes mellitus (DM) [23] and antibiotic use (defined as any use of any antibiotics within 6 months before vaccination) [24]. Overweight/obesity was defined as BMI ≥ 23 kg/m^2^ with reference to National Institutes of Health (NIH) and World Health Organisations (WHO) guidelines for Asians. The correlation between obesity and poor vaccine-induced immune response was observed in hepatitis B [25], tetanus [26], rabies [27] and COVID-19 vaccines [5]. Data also reveal that efficacy of COVID-19 vaccine-induced neutralizing humoral immunity is potentially reduced among the obese subjects (seroconversion rate 82% and 98% in obese and normal BMI subjects, respectively). Poor vaccine-induced antibody protection in obese recipients suggests underlying factors related to obesity limit vaccine response [5]. Diabetes mellitus was defined as hemoglobin A1c ≥ 6.5% or fasting glucose > 7 mmol/L. Immunogenicity of the COVID-19 vaccines has mostly been reported to be lower among patients with DM compared to healthy controls in a recent meta-analysis, regardless of vaccine type [6]. NAFLD is associated with distinct changes in gut microbiota profile [28]. Gut microbiota are important in modulating immune response to different types of vaccination [29,30], including influenza vaccine immunogenicity, which may be affected by antibiotic-induced gut microbiota perturbation [31,32]. Antibiotic-induced gut dysbiosis has also been shown to affect various outcomes, including response to immune checkpoint inhibitors [33] and colorectal cancer development [34].

### 2.4. Statistical Analysis

All statistical analyses were performed using R version 4.1.2 (R Foundation for Statistical Computing, Vienna, Austria) statistical software. The values of continuous variables were displayed as medians and interquartile range (IQR), while values of categorical variables were displayed as numbers and percentages. For two continuous variables, the Mann–Whitney *U*-test was used. For categorical variables, the chi-square test or Fisher exact test was used.

A multivariable logistic regression model was applied to estimate the adjusted odds ratio (aORs) of seroconversion rate and vaccine protection to COVID-19 infection with moderate/severe NAFLD as well as all the aforementioned covariates. An MN titer less than 10 was expressed as 5 for the purpose of statistical analysis.

Sensitivity analysis was performed by reclassifying subjects with mild hepatic steatosis into the NAFLD group.

The statistical significance level threshold was set at *p*-value ≤ 0.05 and all tests were two-sided.

## 3. Results

### 3.1. Demographics and Baseline Characteristics

In total, 259 subjects were enrolled; 68 had moderate (n = 20) or severe (n = 48) hepatic steatosis (NAFLD), and 191 had mild (n = 31) or no (n = 160) hepatic steatosis (control). The demographics of subjects are displayed in Table 1. The median age was similar between NAFLD patients and controls (NAFLD: 51.0 years vs. control: 50.8 years; *p* = 0.271). There were more males in the NAFLD group than controls (52.9% vs. 28.3%; *p* < 0.001). There was a higher proportion of NAFLD patients being overweight or obese compared to controls (89.7% vs. 39.8%; *p* < 0.001). A higher proportion of NAFLD patients had DM compared to controls (17.6% vs. 3.1%; *p* < 0.001).

### 3.2. Comparison of Vaccine Immunogenicity to Wild-type SARS-CoV-2 between NAFLD and Control Groups

Table 2, Appendix A show the seroconversion rate and vMN GMT of the BNT162b2 recipients. At day 21, there was no significant difference in seroconversion rate between NAFLD and control groups (72.1% vs. 77.0%; *p* = 0.418) or the vMN GMT (13.4 vs. 13.6; *p* = 0.885). At day 56, all vaccines achieved seroconversion with a similar vMN GMT (90.4 vs. 99.6; *p* = 0.610). At day 180, more than 97% remained seropositive, with vMN GMT decreasing from 90.4 to 33.3 in NAFLD and from 99.7 to 35.6 in the control group, and there was no significant difference between NAFLD and control groups.

Sensitivity analysis by reclassifying subjects with mild hepatic steatosis into NAFLD group shows similar results (Appendix A).

In univariate analysis, the OR of seropositivity for wild-type virus with male sex was 0.48 (95% CI: 0.27–0.87) (Table 3). Other factors, including age, NAFLD, overweight/obesity, DM and antibiotic use, were not associated with seropositivity to wild-type SARS-CoV-2. In multivariable analysis, male sex remained as the only independent factor with seropositivity (aOR: 0.57, 95% CI: 0.28–0.94) (Table 3).

### 3.3. Comparison of Vaccine Immunogenicity to SARS-CoV-2 Delta/Omicron Variant between NAFLD and Control Groups

There was no significant difference in the seroconversion rate of neutralizing antibodies to the delta variant between NAFLD and control groups, at day 21 (25.0% vs. 29.5%; *p* = 0.70), day 56 (100% vs. 98.4%; *p* = 0.57) and day 180 (89.5% vs. 93.3%; *p* = 0.58), respectively, or, alternatively, the vMN GMT at day 21 (6.83 vs. 6.95; *p* = 0.76), day 56 (62.77 vs. 53.75; *p* = 0.51) and day 180 (20 vs. 25.49; *p* = 0.25), respectively (Table 4, Appendix A).

There was also no significant difference in seroconversion rate of neutralizing antibody to omicron variant between NAFLD and control groups. By day 21, none achieved seroconversion. At day 56, less than 20% achieved seroconversion, and there was no significant difference in seroconversion rate (15.0% vs. 18.0%; *p* = 0.76) or the vMN GMT (5.55 vs. 5.86; *p* = 0.71) among NAFLD and control groups. At day 180, all vaccines became seronegative in both groups.

Sensitivity analysis by reclassifying subjects with mild hepatic steatosis into the NAFLD group shows similar results (Appendix A).

### 3.4. Comparison of Vaccine Protection to SARS-CoV-2 Infection (Any Variants) between NAFLD and Control Groups

There were 4 (1.5%) pieces of missing data on infection rate out of 259 study subjects. Thus, 55 of 255 (21.6%) study subjects had SARS-CoV-2 infection as of 18 May 2022. The median time from vaccination with the first dose to infection was 244 days (IQR: 227.5–264.0). All infections were mild and did not require hospitalization. There was no significant difference in the seroconversion rate at all time points (day 21, day 56 and day 180) between the infected and non-infected subjects (all *p* > 0.05; Appendix A).

There was no significant difference in the infection rate between NAFLD and control groups (25.8% vs. 20.1%; *p* = 0.337). In univariate and multivariable analyses, factors, including age, sex, NAFLD, overweight/obesity, DM, and antibiotic use, were not associated with vaccine protection to SARS-CoV-2 infection (Table 5).

### 3.5. Safety

Thus, 240 (92.7%) BNT162b2 recipients reported adverse effects within 7 days of either the first or the second dose of vaccine (Appendix A). All the adverse effects were mild to moderate (grade 1 and 2) and self-limiting, with no serious adverse events (grade 3 and 4), such as anaphylaxis or cardiovascular events. The most common local adverse events were injection site pain (88.9%), while the most common systemic adverse reaction was fatigue (52.5%). Overall, the rate of adverse events was similar between NAFLD and control groups (63 (92.6%) vs. 177 (92.7%), *p* = 0.343). Among systemic adverse reactions, the NAFLD group showed a higher rate of chills and rigors (12 (17.6%) vs. 21 (11.0%), *p* = 0.042), joint pain (12 (17.6%) vs. 27 (14.1%), *p* = 0.026) and nausea (6 (8.8%) vs. 13 (6.8%), *p* = 0.027) than the control group. There was no significant difference in the rate of local adverse reactions between NAFLD and control groups (59 (86.8%) vs. 173 (90.6%), *p* = 0.412).

## 4. Discussion

This prospective cohort study demonstrates that there was no difference in the vaccine efficacy in terms of neutralizing antibody response to wild-type and mutant SARS-CoV-2 between moderate to severe NAFLD and control groups. The seroconversion rate of neutralizing antibodies against the wild-type, delta variant and omicron variant was >97%, >89% and 0% after 6 months, respectively. There was also no difference in the rate of COVID-19 infection between the two groups (25.8% vs. 20.1%).

It has been observed that COVID-19 patients with chronic liver disease had increased length of hospital stay, higher rates of intensive care unit stay and need for mechanical ventilation compared to those without chronic liver disease. This association was also observed in patients with NAFLD, even after controlling for the presence of obesity [35]. The effect of NAFLD on COVID-19 severity may be due to underlying obesity and hepatic steatosis with higher serum markers of inflammation and oxidative stress [36]. Vaccination is paramount in preventing SARS-CoV-2 infection, severe symptoms and death. There are few studies that have evaluated the efficacy of SARS-CoV-2 vaccination in patients with NAFLD, and available studies are limited to assessing vaccine immunogenicity to wild-type SARS-CoV-2 infection only.

As far as we know, our study is the first to compare the immunogenicity of mRNA vaccines to different strains of COVID-19, including wild-type, delta and omicron variants, between NAFLD and control groups. Another merit of this study was the prolonged follow-up to more than 6 months in terms of immunogenicity and infection outcome. In a multicenter study conducted in China, Wang et al. [37] reported an encouraging result of more than 95% of NAFLD patients elicited detectable neutralizing antibody responses after two doses of the inactivated COIVD-19 vaccine (BBIBP-CorV). However, the vaccine studied (BBIBP-CorV) was an inactivated vaccine and there was no comparison with a control group. Moreover, status of NAFLD might be misclassified as the diagnosis of NAFLD was heterogeneously defined by either clinical findings or liver biopsy.

Our study had additional advantages. First, live virus, the gold standard for analysis of vaccine humoral response [38], was used, in comparison with a surrogate virus neutralization test, where correlation with live virus was only 0.7–0.8 [39]. Second, a homogeneous definition of NAFLD using CAP measurement from transient elastography was adopted, which also allowed us to analyze vaccine immunogenicity based on the severity of NAFLD.

A prospective cohort study [14] demonstrated that a lower proportion of moderate or severe hepatic steatosis patients, as compared to the control group, achieved the highest-tier response for either mRNA (BNT162b2) or inactivated vaccines (CoronaVac). However, SARS-CoV-2 variants were not evaluated and vaccine immunogenicity on day 180 was not well studied due to the relative proportion of missing data. Our current study emphasized the long-term immunogenicity, as well as a comparison of, SARS-CoV-2 variants (delta variants and omicron variants).

We found that BNT162b2 was effective against wild-type SARS-CoV-2, but there was no difference in the seroconversion rate or vMN GMT after either the first or second dose between the NAFLD and control groups at different time points. At day 21, more than 70% achieved seropositivity; at day 56, all vaccinees attained seroconversion with a similar vMN GMT (90.4 vs. 99.7, *p* = 0.61); and at day 180, over 97% remained seropositive with a similar vMN GMT.

Multivariable analysis further shows that male sex was the only independent factor predicting serological response to wild-type SARS-CoV-2 but not other factors, including NAFLD and cardiovascular risk factors. Similar findings were observed for SARS-CoV-2 delta and omicron variants, in which there was no significant difference in the seroconversion rate of neutralizing antibody nor vMN GMT to SARS-CoV-2 delta and omicron variant among the NAFLD and control group. Of note, BNT162b2 vaccine efficacy remained high to delta variants, in which more than 98% achieved seropositivity at day 56. However, BNT162b2 vaccine was less effective on omicron variants, in which less than 20% achieved seroconversion at day 56. A similar proportion of NAFLD and control groups (25.8% vs. 20.1%) had COVID-19 disease despite receiving two doses of BNT162b2. All infections were contracted after day 180 of first dose BNT162b2 vaccine, in early 2022, during which the omicron variant was the predominant strain in Hong Kong.

With regard to safety, COVID-19 vaccines were well tolerated, with mild and self-limiting side effects. They were generally similar between NAFLD and control groups in terms of overall adverse events. It was observed that the NAFLD group reported a higher frequency of chills and rigors, nausea and joint pain than control group. Nonetheless, the symptoms were mild and resolved within days. The association between reactogenicity and immunogenicity is still not well established and yet to be answered.

There were several limitations in our study. First, the sample size was relatively small, and this study was only limited to the BNT162b2 vaccine. Future studies with a larger sample size and assessment of different COVID-19 vaccine platforms, in particular to the latest COVID-19 bivalent vaccine boosters, will allow for better evaluation. Second, vaccine-induced cellular immunity against SARS-CoV-2 was not studied. Given the role of CD4 and CD8 T cells in the clearance of infections, via suppression of viral replication and mounting of long-term memory of the immune system, it was believed that vaccine-induced T-cell response may substantially protect against severe SARS-CoV-2 disease, even with antibody seronegativity [40]. This may be specifically relevant for the omicron variant, which dramatically evades neutralizing antibody responses [41].

Third, NAFLD was defined by measurement of CAP using transient elastography. However, magnetic resonance elastography (MRE), which has a diagnostic accuracy of 0.9, remains as the most accurate method in diagnosing NAFLD among the available non-invasive modalities. Fourth, more long-term data on immunogenicity data beyond 180 days are lacking. With the advocation of third, fourth and even fifth dose of vaccine, longer-term follow-up (e.g., one year) and further investigation on serological response to additional doses of vaccine in NAFLD patients are needed.

There are two implications in our study. First, given the good immunogenicity of the BNT162b2 vaccine, with comparable effectiveness on COVID-19 protection between NAFLD and control groups, NAFLD patients should be ascertained and encouraged to receive vaccination (at least two doses) to prevent severe complications. Second, the BNT162b2 vaccine is proven to be safe. Mild and self-limiting side effects should not deter NAFLD patients from being vaccinated in exchange for immunogenicity to SARS-CoV-2.

## 5. Conclusions

There was no difference in the seroconversion rate to wild-type SARS-CoV-2 and variants between moderate to severe NAFLD and control groups after two doses of BNT162b2. The BNT162b2 vaccine had good immunogenicity to the wild-type and delta variants but not the omicron variant in patients with NAFLD.

## Figures and Tables

**Table 1 vaccines-11-00497-t001:** Baseline characteristics of BNT162b2 recipients.

	Whole Cohort (n = 259)	NAFLD (n = 68)	Control (n = 191)	*p*-Value
Age (years)	50.8 (43.6–57.8)	51.0 (46.2–57.7)	50.8 (40.9–57.8)	0.271
Male sex (n, %)	90 (34.7)	36 (52.9)	54 (28.3)	<0.001
Overweight/obesity (n, %) (BMI ≥ 23 kg/m^2^)	137 (52.9) *	61 (89.7)	76 (39.8) *	<0.001
Diabetes mellitus (n, %)	18 (6.9)	12 (17.6)	6 (3.1)	<0.001
Antibiotic use (n, %)	20 (7.7)	6 (8.8)	14 (7.3)	0.692
Liver stiffness (kPa)	4.5 (3.7–5.3)	4.8 (4.12–5.9)	4.3 (3.6–5.1)	<0.001
CAP score (dB/m)	230 (203–269)	294 (276–312.5)	216 (196.5–236.5)	<0.001

Note: Data are displayed as median (interquartile range) and number (%). Abbreviations: NAFLD, non-alcoholic fatty liver disease; BMI, body mass index; CAP controlled attenuation parameter. * 1 missing data.

**Table 2 vaccines-11-00497-t002:** Antibody responses to wild-type SARS-CoV-2 among BNT162b2 recipients (n = 259).

	NAFLD (n = 68)	Control (n = 191)	*p*-Value
Seroconversion rate *			
D21	49/68 (72.1)	147/191 (77.0)	0.418
D56	68/68 (100)	189/189 (100) ^	1
D180	53/53 (100) ^	140/144 (97.2) ^	0.220
vMN GMT			
D21	13.44 (11.02–16.44)	13.56 (12.18–15.18)	0.885
D56	90.41 (75.19–108.85)	99.69 (88.23–112.17)	0.610
D180	33.31 (27.11–40.85)	35.64 (31.19–40.85)	0.691

Note: Data are displayed as median (interquartile range) and number (%), Abbreviations: NAFLD, non-alcoholic fatty liver disease; D21 day 21; D56 day 56; D180 day 180. * Seroconversion rate was considered as positive if MN titre ≥ 10. ^ data not available were excluded.

**Table 3 vaccines-11-00497-t003:** Factors associated with serological response to wild-type SARS-CoV-2 at day 21 among BNT162b2 recipients in multivariable analysis.

	Univariate Analysis Odds Ratio	*p*-Value	Multivariable Analysis Adjusted Odds Ratio	*p*-Value
Age (≥60 years)	0.56 (0.28–1.14)	0.101	0.57 (0.28–1.19)	0.126
Male sex	0.48 (0.27–0.87)	0.015	0.57 (0.28–0.94)	0.032
NAFLD (CAP ≥ 268 dB/m)				
	0.77 (0.42–1.47)	0.419	1.03 (0.50–2.17)	0.934
DM	0.82 (0.30–2.66)	0.724	1.22 (0.41–4.18)	0.727
Overweight/Obesity (BMI ≥ 23 kg/m^2^)				
	0.67 (0.37–1.19)	0.176	0.75 (0.39–1.45)	0.391
Antibiotic use	0.96 (0.36–3.06)	0.942	0.87 (0.31–2.80)	0.793

Abbreviations: NAFLD, non-alcoholic fatty liver disease; CAP, controlled attenuation parameter; DM, diabetes mellitus; BMI, body mass index.

**Table 4 vaccines-11-00497-t004:** Antibody responses to SARS-CoV-2 delta and omicron variant among BNT162b2 recipients (n = 81).

	NAFLD (n = 20)	Control (n = 61)	*p*-Value
**Delta variant**			
Seroconversion rate *			
D21	5/20 (25.0)	18/61 (29.5)	0.698
D56	20/20 (100)	60/61 (98.4)	0.565
D180	17/19 (89.5) ^	56/60 (93.3) ^	0.580
vMN GMT			
D21	6.83 (5.10–9.12)	6.95 (6.05–8.00)	0.761
D56	62.77 (44.26–89.12)	53.75 (43.82–66.02)	0.507
D180	20.00 (13.87–29.08)	25.49 (20.70–31.50)	0.247
**Omicron variant**			
Seroconversion rate *			
D21	0/20 (0)	0/61 (0)	1
D56	3/20 (15.0)	11/61 (18.0)	0.756
D180	0/19 (0) ^	0/60 (0) ^	1
vMN GMT			
D21	UD (UD)	UD (UD)	1
D56	UD (UD)	UD (UD)	1
D180	UD (UD)	UD (UD)	1

Note: Data are displayed as median (interquartile range) and number (%). Abbreviations: NAFLD, non-alcoholic fatty liver disease. * Seroconversion rate was considered as positive if MN titre ≥ 10. ^ data not available were excluded.

**Table 5 vaccines-11-00497-t005:** Factors associated with SARS-CoV-2 infection among BNT162b2 recipients on multivariable analysis.

	Univariate Analysis Odds Ratio	*p*-Value	Multivariable Analysis Adjusted Odds Ratio	*p*-Value
Age (≥60 years)	0.52 (0.19–1.23)	0.165	0.50 (0.18–1.21)	0.151
Male sex	0.59 (0.29–1.14)	0.129	0.52 (0.25–1.04)	0.074
NAFLD (CAP ≥ 268 dB/m)	1.38 (0.70–2.63)	0.338	1.50 (0.68–3.24)	0.308
DM	1.44 (0.44–4.02)	0.508	1.71 (0.49–5.32)	0.372
Overweight/Obesity (BMI ≥ 23 kg/m^2^)	1.11 (0.61–2.03)	0.738	1.02 (0.51–2.03)	0.948
Antibiotic use	1.23 (0.39–3.36)	0.698	1.16 (0.36–3.22)	0.787

Abbreviations: NAFLD, non-alcoholic fatty liver disease; CAP, controlled attenuation parameter; DM, diabetes mellitus; BMI: body mass index.

## Data Availability

Data will be shared upon reasonable request.

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
