# Peer review of "Effect of Moderate to Severe Hepatic Steatosis on Vaccine Immunogenicity against Wild-Type and Mutant Virus and COVID-19 Infection among BNT162b2 Recipients"

_vaccines, 2023, doi:10.3390/vaccines11030497_

Round 1

Reviewer 1 Report

The present study addresses the impact of NAFLD upon the BioNTech-Pfizer vaccine efficiency in terms of seroconversion.

I fully agree with the authors that viral neutralization tests are to be considered the gold standard in terms of evaluating disease protection.

Fibroscan was used to evaluate the hepatic steatosis – this is again a correct approach. The limits between groups were considered according to literature. However, could you please offer details regarding the distribution/number of patients/healthy individuals into the normal, mild, moderate and severe steatosis?

Lines 44-45: please insert some references

Furthermore, given the topic, additional literature data regarding the impact of risk factors such as diabetes, obesity etc upon the efficacy of COVID-19 vaccines (mostly Pfizer) is mandatory.

Lines 54-56: I must admit I have failed to understand what is the connection between the present study and the one of Wang et al. regarding the SINOPHARM vaccine in non NAFLD patients.

Lines 88-90: please revise English-wise

Line 93: please explain TCID50

Lines 110-111: Please revise English-wise. Also, please explain again in this paragraph why was this test performed. The sentence starting with “Subjects were requested …” has no connection with the previous one and should not only be placed in a different paragraph, but also should be properly introduced.

Line 119: measured

Lines 128-130:  Such correlations are cited for the anti-COVIF-19 vaccines as well. Please insert

Lines 139-140: As much as this statement is correct, there is no mentioned link between NAFLD and microbiota perturbation. Furthermore, antibiotic treatment was not considered as an exclusion factor, hence this paragraph requires adequate editing.

One of your conclusions states that the male sex is an independent factor predicting serological conversion. However, Table 1 shows that 90% of your subjects are males. Please explain.

Lines 209-210:  It would be very helpful to show the distribution of the 55 infected patients within the NFALD vs control groups. Were there any differences in terms of seroconversion between these infected patients and the others at various time points?

Line 211: vaccination with

Lines 291-294: I agree with the idea that there is no clear association between reactogenicity and immunogenicity. You mention the higher frequency of side effects in NFALD patients. Such symptoms might actually suggest, paradoxically, a stronger, more vigorous immune response, so this offers ground for further speculations and perhaps a future direction of research.

I also agree with your comments regarding the study limitations. In fact, the immune status of your patients was not at all assessed, and this should be considered as a major drawback. It is acknowledged that the mRNA-based vaccines induce a Th1 response, hence the class switch bias.

Author Response

16 February 2023

Dear Prof Tripp

Title: Effect of moderate-to-severe hepatic steatosis on vaccine immunogenicity against wild-type and mutant virus and COVID-19 infection among BNT162b2 recipients 

Thank you for your letter concerning the revision of our manuscript.  The manuscript has been revised according to the Editorial Board and reviewer’s comments. The changes have been highlighted in the manuscript for your reference. The followings are the point-by-point responses to the reviewers’ comments.

We hope that the reviewers and editors will find this revised version acceptable for publication in Vaccines.

Yours sincerely,

Man Fung Yuen

Reviewer 1

Top of Form

Comments and Suggestions for Authors

The present study addresses the impact of NAFLD upon the BioNTech-Pfizer vaccine efficiency in terms of seroconversion.

I fully agree with the authors that viral neutralization tests are to be considered the gold standard in terms of evaluating disease protection.

Fibroscan was used to evaluate the hepatic steatosis – this is again a correct approach. The limits between groups were considered according to literature. However, could you please offer details regarding the distribution/number of patients/healthy individuals into the normal, mild, moderate and severe steatosis?

We thank you for your comments.

With regard to the details of the distribution if each subcategory, the number of patients in normal, mild, moderate and severe steatosis would be 160, 31, 20 and 48 respectively (page 4 lines 205-206).

Subjects with moderate/severe HS have markedly higher risks than those with mild HS in various clinical outcomes including fibrosis/cirrhosis, HCC, and cardiovascular diseases (e.g. myocardial infarction). (page 3 lines 139-141).  

Therefore, we would like to classify patient into two groups as in our study design, no/mild HS and moderate/severe HS.

Lines 44-45: please insert some references

Furthermore, given the topic, additional literature data regarding the impact of risk factors such as diabetes, obesity etc upon the efficacy of COVID-19 vaccines (mostly Pfizer) is mandatory.

We thank you for your comments

We have added the references on the impact of risk factors on comorbidities on COVID-19 outcomes and vaccine immunogenicity (page 2 lines 44-46)

Lines 54-56: I must admit I have failed to understand what is the connection between the present study and the one of Wang et al. regarding the SINOPHARM vaccine in non NAFLD patients.

The difference between Wang’s study and ours (page 2 lines 58-62)

  1. BBIBP-CorV (inactivated vaccine) vs BNT162b2 (mRNA vaccine)
  2. Wang did not recruit patients without NAFLD for comparison
  3. Wang did not report the long term immunogenicity at 6 months
  4. Wang did not report immunogenicity against mutant viruses
  5. Wang did not report infection outcome

Therefore, we aimed to address the limitations of prior study

Lines 88-90: please revise English-wise

We thank you for your comments. We have revised it as “VNTs, which are strongly correlated with disease protection, was chosen as the indicator of COVID-vaccine efficacy in our study”. (page 2 lines 90-91).

Line 93: please explain TCID50

We thank you for your comments. TCID50 refers to 50% tissue culture infective dose (page 2 line 94). It is defined as the dilution of a virus required to infect 50% of a given cell culture. It is used to quantify and to assess the infectivity of a virus in cells. 

Lines 110-111: Please revise English-wise. Also, please explain again in this paragraph why was this test performed. The sentence starting with “Subjects were requested …” has no connection with the previous one and should not only be placed in a different paragraph, but also should be properly introduced.

We thank you for your comments. Revision has been made as follows: For the outcome of infection, subjects were followed till 18th May 2022 (study end date). COVID-19 infection was confirmed by either Rapid Antigen test (RAT) or Deep Throat Saliva (DTS). For the outcome of adverse reactions, subjects were requested to report any adverse reactions daily for 7 days after each dose of vaccine (page 3 lines 122-125).

Line 119: measured

We thank you for your comments. Revision has been made (page 3 line 133).

Lines 128-130:  Such correlations are cited for the anti-COVIF-19 vaccines as well. Please insert

We thank you for your comments. Revision has been made (page 3 line 147).

Lines 139-140: As much as this statement is correct, there is no mentioned link between NAFLD and microbiota perturbation. Furthermore, antibiotic treatment was not considered as an exclusion factor, hence this paragraph requires adequate editing.

One of your conclusions states that the male sex is an independent factor predicting serological conversion. However, Table 1 shows that 90% of your subjects are males. Please explain.

We thank you for your comments. We have added the sentence “NAFLD is associated with distinct changes of gut microbiota profile” (page 3 lines 154-155). Although antibiotic treatment was not an exclusion factors, it was adjusted for in the multivariable analysis (page 4 line 197).

Both the number and proportion of male subjects are stated in Table 1.  Among the whole cohort, 90 out of 259 are male, that accounts for 34.7%.

Lines 209-210:  It would be very helpful to show the distribution of the 55 infected patients within the NFALD vs control groups. Were there any differences in terms of seroconversion between these infected patients and the others at various time points?

Table 2

We thank you for your comments.

There were two missing data in NAFLD group and two missing data in control group.

A similar proportion of NAFLD and control groups (17 out of 66 [25.8%] vs 38 out of 189 [20.1%]) had COVID-19 disease despite receiving two doses of BNT162b2 (page 7 lines 290-291).

There was no significant difference in the seroconversion rate at all time points (day 21, day 56 and day 180) between the infected and non-infected subjects (all p > 0.05; page 7 lines 286-288, eTable 2 newly added)

Line 211: vaccination with

We thank you for your comments. Revision has been made (page 7 line 285).

Lines 291-294: I agree with the idea that there is no clear association between reactogenicity and immunogenicity. You mention the higher frequency of side effects in NFALD patients. Such symptoms might actually suggest, paradoxically, a stronger, more vigorous immune response, so this offers ground for further speculations and perhaps a future direction of research.

I also agree with your comments regarding the study limitations. In fact, the immune status of your patients was not at all assessed, and this should be considered as a major drawback. It is acknowledged that the mRNA-based vaccines induce a Th1 response, hence the class switch bias.

We thank you for your comments. We acknowledged the limitation of our study that the role of CD4 and CD8 were not examined. This would be a future direction of research where vaccine-induced T-cell response should be determined and thus addressed the missing gap of current studies. (page 9 lines 389-395).

Bottom of Form

Reviewer 2 Report

In this modest prospective cohort study in 259 recipients of the BNT162b2 (mRNA) vaccine for COVID-19, the authors have compared the immunogenicity of the mRNA vaccine in individuals with fatty liver disease 15 (NAFLD) to the corresponding immunogenicity in mild hepatic steatosis and healthy individuals (controls). Their data demonstrate statistically equivalent virus neutralization seroconversion rates and titres against the ancestral and two variants of SARS-CoV-2 in individuals with moderate to severe NAFLD compared to or those with mild NAFLD up to 6 months after two doses of the vaccine. They conclude that NAFLD patients who receive the initially recommended two-dose vaccination two-dose schedule of the mRNA vaccine are ‘not at higher risk of infection than healthy individuals. Multivariable logistic regression analysis with several COVID-19-related covariates did not identify any factors affecting immunogenicity of this vaccine in NAFLD patients.

A prospective study design and carefully conducted study supports the authors’ conclusions with respect to immunogenicity of the vaccine. However, I advise caution in interpreting virus neutralisation titres (VNTs) as evidence of vaccine effectiveness or protection from infection (see comments below).  Numerous studies have shown that the mRNA vaccines do not protect from infection, but do prevent severe Infections. It is not clear why mild NAFLD patients were included in the control group and sensitivity analysis, for example, re-classifying mild disease could help to determine the contribution of this subgroup to the conclusions. With the current recommendations for additional (‘booster’) doses of COVID-19 vaccines, the effect of boosting with a third dose in special populations like this cohort would be of great interest.

Specific questions and suggestions

Line 71-72: The authors should clarify whether all participants were tested for infections using the N-protien antibody detection or if some were diagnosed by history alone.

Line 88-89: While VNTs do, indeed, correlate with protection for severe infections, vaccine effectiveness involves protection from infection irrespective of severity. “Efficacy’ may be a more appropriate term.

Line 111: What is the significance of the date May 18, 2022?

Line 124: Why were participants with mild steatosis grouped with healthy participants?  Does. Re-classifying them as NAFLD change the outcome of analyses?

Table 4: Which substrain of Omicron was tested in the VNTs against variants?

Author Response

16 February 2023

Dear Prof Tripp

Title: Effect of moderate-to-severe hepatic steatosis on vaccine immunogenicity against wild-type and mutant virus and COVID-19 infection among BNT162b2 recipients 

Thank you for your letter concerning the revision of our manuscript.  The manuscript has been revised according to the Editorial Board and reviewer’s comments. The changes have been highlighted in the manuscript for your reference. The followings are the point-by-point responses to the reviewers’ comments.

We hope that the reviewers and editors will find this revised version acceptable for publication in Vaccines.

Yours sincerely,

Man Fung Yuen

Reviewer 2

In this modest prospective cohort study in 259 recipients of the BNT162b2 (mRNA) vaccine for COVID-19, the authors have compared the immunogenicity of the mRNA vaccine in individuals with fatty liver disease 15 (NAFLD) to the corresponding immunogenicity in mild hepatic steatosis and healthy individuals (controls). Their data demonstrate statistically equivalent virus neutralization seroconversion rates and titres against the ancestral and two variants of SARS-CoV-2 in individuals with moderate to severe NAFLD compared to or those with mild NAFLD up to 6 months after two doses of the vaccine. They conclude that NAFLD patients who receive the initially recommended two-dose vaccination two-dose schedule of the mRNA vaccine are ‘not at higher risk of infection than healthy individuals. Multivariable logistic regression analysis with several COVID-19-related covariates did not identify any factors affecting immunogenicity of this vaccine in NAFLD patients.

A prospective study design and carefully conducted study supports the authors’ conclusions with respect to immunogenicity of the vaccine. However, I advise caution in interpreting virus neutralisation titres (VNTs) as evidence of vaccine effectiveness or protection from infection (see comments below).  Numerous studies have shown that the mRNA vaccines do not protect from infection, but do prevent severe Infections. It is not clear why mild NAFLD patients were included in the control group and sensitivity analysis, for example, re-classifying mild disease could help to determine the contribution of this subgroup to the conclusions. With the current recommendations for additional (‘booster’) doses of COVID-19 vaccines, the effect of boosting with a third dose in special populations like this cohort would be of great interest.

Specific questions and suggestions

Line 71-72: The authors should clarify whether all participants were tested for infections using the N-protien antibody detection or if some were diagnosed by history alone.

We thank you for your comments.

Prior COVID-19 infection was identified from both history taking and testing for antibodies to SARS-CoV-2 nucleocapsid (N) protein (page 2 line 72-73).

Line 88-89: While VNTs do, indeed, correlate with protection for severe infections, vaccine effectiveness involves protection from infection irrespective of severity. “Efficacy’ may be a more appropriate term.

We thank you for your comments. Revision has been made (page 2 line 73).

Line 111: What is the significance of the date May 18, 2022?

We thank you for your comments. This is the study end date for investigating the infection outcome (page 3 line 123).

Line 124: Why were participants with mild steatosis grouped with healthy participants?  Does. Re-classifying them as NAFLD change the outcome of analyses?

We thank you for your comments.

Subjects with moderate/severe HS have markedly higher risks than those with mild HS in various clinical outcomes including fibrosis/cirrhosis, HCC, and cardiovascular diseases (e.g. myocardial infarction). (page 3 lines 139-141).  

Therefore, we would like to classify patient into two groups as in our study design, no/mild HS and moderate/severe HS.

In addition, we have conducted a sensitivity analysis as suggested by reclassifying mild hepatic steatosis subjects into the NAFLD group (page 4 lines 199-200). There remains no significant difference in the seroconversion rate and vMN GMT for wild type or mutants between the newly defined NAFLD and control groups (page 5 lines 231-232 and page 6 lines 269-270, eTable 1 newly added).

Table 4: Which substrain of Omicron was tested in the VNTs against variants?

We thank you for your comments. It is BA.1 variant (page 3 line 113).